# DEFENDING DNN ADVERSARIAL ATTACKS WITH PRUNING AND LOGITS AUGMENTATION

**Shaokai Ye[1*], Siyue Wang[2*], Xiao Wang[3], Bo Yuan[4], Wujie Wen[5] & Xue Lin[2]**
1. Syracuse University 2. Northeastern University 3. Boston University
4. City University of New York 5. Florida International University          [*]Equal Contribution
sy106@syr.edu wang.siy@husky.neu.edu kxw@bu.edu byuan@ccny.cuny.edu
wwen@fiu.edu xue.lin@northeastern.edu

## ABSTRACT

Deep neural networks (DNNs) have been shown to be powerful models and perform extremely well on many complicated artificial intelligent tasks. However, recent research found that these powerful models are vulnerable to adversarial attacks, i.e., intentionally added imperceptible perturbations to DNN inputs can easily mislead the DNNs with extremely high confidence. In this work, we enhance the robustness of DNNs under adversarial attacks by using pruning method and logits augmentation, therefore, we achieve both higher defense against adversarial examples and more compressed DNN models. We have observed defense against adversarial attacks under the white box attack assumption. Our defense mechanisms work even better under the black box attack assumption.

## 1 INTRODUCTION

Deep neural networks (DNNs) are powerful models that achieve state-of-the-art performance in various speech and visual recognition tasks, including speech recognition, natural language processing, scene understanding, object recognition, etc. As a key enabler of DNNs, the large model size also demands increasing computation and memory resources from the computing platforms. It has been investigated that DNNs are robust to random perturbations (Fawzi et al., 2016). However, recent study (Szegedy et al., 2013), Goodfellow et al. (2014), Kurakin et al. (2016) shows that DNNs are vulnerable to adversarial attacks, that is, intentionally added imperceptible perturbations to DNN inputs can easily mislead the DNNs with extremely high confidence.

Goodfellow et al. (2014), Kurakin et al. (2016), Papernot et al. (2016b) and Carlini & Wagner (2017) have implemented the adversarial attacks by generating adversarial examples. If the neural network classifies a legal input $x$ with label $C$, an adversarial example $x'$ is the one that very similar to $x$ according to some distance metrics and will be labeled by the neural network as $C' \neq C$. Goodfellow et al. (2014) proposed Fast Gradient Sign Method (FGSM) that uses the gradient of the loss function to determine the direction of the perturbation and sets a constant for the intensity of the perturbation. This method features very fast speed for generating adversarial examples. Kurakin et al. (2016) enhanced the FGSM by taking multiple smaller steps in the direction of gradient sign, which is known as Basic Iterative Method (BIM). Carlini & Wagner (2017) claimed to generate the strongest attacks (the C&W method) by solving an optimization problem with minimizing some distance metrics, i.e., $L_p$ norm, where $p = 0, 2$, and $\infty$ are used. C&W is the strongest attacks in that it conquered multiple defense methods in the *white box* assumption, i.e., the attacker has perfect knowledge about the targeted DNN model and data set.

Previous works have been done on improving the robustness of DNNs under adversarial attacks. Papernot et al. (2016c) proposed their defensive mechanism called defensive distillation by modifying the model parameters (softmax function) in order to increase the robustness of DNNs. Others (Feinman et al., 2017; Bhagoji et al., 2017) tried to defend the adversarial examples by attempting to detect them. Feinman et al. (2017), through modeling the outputs from the final hidden layer of DNNs, indicated that there exists the difference in the distribution of adversarial examples and legal examples, while Bhagoji et al. (2017) proposed a defense mechanism based on dimensionality reduction.

Table 1: Adversarial attack successful rates of the unprotected model M0, Level One model M1, and Level Two model M2 under four attacks (untargeted FGSM, targeted FGSM, targeted BIM, and C&W) using MNIST dataset.

| Attack Method | Untargeted FGSM | | | Targeted FGSM | | | Targeted BIM | | | C&W |
|---|---|---|---|---|---|---|---|---|---|---|
| Parameters | $\varepsilon = 0.1$ | $\varepsilon = 0.15$ | $\varepsilon = 0.25$ | $\varepsilon = 0.1$ | $\varepsilon = 0.15$ | $\varepsilon = 0.25$ | $\varepsilon = 0.1$ | $\varepsilon = 0.15$ | $\varepsilon = 0.25$ | iter = 100 |
| M0 | 9.0% | 17.0% | 45.6% | 1.97% | 4.52% | 12.0% | 3.89% | 14.81% | 39.64% | 99.6% |
| Distortion | (2.19) | (3.28) | (5.45) | (2.17) | (3.25) | (5.39) | (2.11) | (3.11) | (5.28) | (2.03) |
| M1 | 7.4% | 8.7% | 20.2% | 1.17% | 1.68% | 4.04% | 3.14% | 9.9% | 31.26% | 96.97% |
| Distortion | (2.16) | (3.25) | (5.38) | (2.15) | (3.22) | (5.35) | (2.14) | (3.13) | (5.07) | (2.28) |
| M2 | 1.1% | 1.1% | 1.1% | 1.04% | 1.5% | 3.87% | 2.71% | 7.9% | 21.12% | 95.93% |
| Distortion | (2.28) | (3.41) | (5.65) | (2.15) | (3.22) | (5.35) | (2.15) | (3.1) | (5.1) | (2.5) |

The experiment is evaluated on 1000 source samples from MNIST. We set the search step for line search in C&W as 10.

In this work, we enhance the robustness of DNNs under adversarial attacks by using pruning method and logits augmentation, therefore, we achieve both higher defense against adversarial examples and more compressed DNN models. We have observed defense against adversarial attacks under the white box attack assumption. Our defense mechanisms work even better under the black box attack assumption.

## 2 METHODOLOGY

### 2.1 MODEL COMPRESSION USING PRUNING

To reduce model size and facilitate implementing DNNs for consumer applications, Han et al. (2015) proposed the DNN pruning method that reduces the number of weights while preserving the accuracy of the compressed DNN models. The pruning process starts from learning the connectivity through normal network training, followed by pruning the connections whose weights are below a given threshold. After making it a sparser network, the DNN is retrained to finalize weights of the remaining connections. This pruning-and-retraining process is performed iteratively until the network is pruned to the largest extent without accuracy loss.

In this work, we use a network structure with 4 convolutional layers and 3 fully connected layers. In each iteration, we prune 10% nonzero weights for fully connected layers and 5% nonzero weights for convolutional layers. We prune and train for 20 iterations maintaining the accuracy and the network model can be compressed by $7\times$. We demonstrate in the later Section that the pruning-based model compression method can defend the adversarial attacks, that is, by using pruning method we can achieve both compressed network model size and defense against adversarial attacks.

### 2.2 LOGITS AUGMENTATION

To further improve the robustness of DNNs under adversarial attacks, we propose to use the logits augmentation on top of the pruning method. Inspired by the gradient inhibition method (Liu et al., 2018), which changes the weights in the last few layers as

$$w = w + \tau * sign(w). \tag{1}$$

In our logits augmentation, we modify the weights in the last fully connected layer by

$$w = \tau \times w \tag{2}$$

In our experiments, we set the value of $\tau$ to fine-tune the defense effectiveness. Through a thorough analysis, we find that both the pruning and the logit augmentation can change the distribution of weights in a DNN and therefore achieving some level of defense against adversarial examples.

Table 2: Adversarial attack successful rates of the unprotected model C0, Level One model C1, and Level Two model C2 under four attacks using CIFAR-10 dataset.

| Attack Method | Untargeted FGSM | | | Targeted FGSM | | | Targeted BIM | | | C&W |
|---|---|---|---|---|---|---|---|---|---|---|
| Parameters | $\varepsilon=$ 0.1 | $\varepsilon =$ 0.15 | $\varepsilon =$ 0.25 | $\varepsilon =$ 0.1 | $\varepsilon=$ 0.15 | $\varepsilon =$ 0.25 | $\varepsilon=$ 0.1 | $\varepsilon =$ 0.15 | $\varepsilon =$ 0.25 | iter = 100 |
| C0 | 84.6% | 86.3% | 87.1% | 17.71% | 14.78% | 11.49% | 63.59% | 65.83% | 65.73% | 99.54% |
| Distortion | (5.43) | (8.05) | (13.0) | (5.43) | (8.05) | (13.0) | (4.48) | (6.66) | (10.8) | (2.06) |
| C1 | 70.3% | 75.3% | 80.9% | 11.2% | 10.5% | 10.1% | 25.3% | 23.8% | 19.3% | 85.0% |
| Distortion | (5.43) | (8.05) | (13.0) | (5.42) | (8.05) | (13.03) | (4.47) | (6.64) | (10.8) | (3.55) |
| C2 | 24.6% | 24.5% | 25% | 11.12% | 11.25% | 11.16% | 43.41% | 44.9% | 41.2% | 83.9% |
| Distortion | (1.42) | (2.11) | (3.41) | (5.33) | (7.91) | (12.8) | (4.43) | (6.5) | (10.7) | (4.31) |

The experiment is evaluated on 1000 source samples from CIFAR-10. We set the search step for line search in C&W as 10.

## 3 EXPERIMENTS AND RESULTS

### 3.1 EXPERIMENT SETUP

In order to test our defense mechanisms against adversarial examples, we use three adversarial example generating methods i.e., FGSM by Goodfellow et al. (2014), BIM by Kurakin et al. (2016), and C&W by Carlini & Wagner (2017). We use both MNIST Yann et al. (1998) and CIFAR-10 Krizhevsky & Hinton (2009) datasets to train the network models. The three attacks have been implemented in Cleverhans (Papernot et al., 2016a), a Python library to benchmark machine learning systems' vulnerability to adversarial examples, and we use those source codes directly for generating adversarial examples. We are using the *white box* assumption when generating the adversarial examples, i.e., the attackers have perfect knowledge of all the targeted neural network models, and training and testing datasets.

We start from training unprotected neural network models i.e., M0 and C0, achieving near state-of-the-art accuracies, i.e., 99.4% and 80%, respectively, on MNIST and CIFAR-10 datasets. For defense mechanisms, we test two defending levels: Level One exploits the pruning method Han et al. (2015) only as defense, while Level Two exploits both pruning and logits augmentation as defense. We have M1 and C1 network models for Level One defense, respectively, for the MNIST and CIFAR-10 datasets. We have M2 and C2 network models for Level Two defense. Please note that we do not lose any test accuracy under Level One and Level Two models.

### 3.2 EXPERIMENT RESULTS

Table 1 and Table 2 demonstrate the adversarial attack successful rates of unprotected models, Level One models, and Level Two models under four attacks (untargeted FGSM, targeted FGSM, targeted BIM, and C&W) on MNIST and CIFAR-10 datasets, respectively. We also report with the attack successful rate the aevarge distortion of the adversarial examples compared to the legal examples. The $\varepsilon$ parameter is related to the distortion. When a larger $\varepsilon$ value is used, the resulted distortion is higher. For the untarged FGSM, the adverserial attack successful rate for MNIST dataset (the fourth colomn in Table 1) is reduced from 45.6% by unprotected model M0 to 20.2% by Level One model M1 and to 1.1% by Level Two model M2. Similarly, we can observe decrease in the adversarial example successful rate under other attacks and for CIFAR-10 dataset. Here we find that our defense mechanisms do not defend the C&W attacks as much as we do for the other attacks under the pure white box assumption.

Furthermore, we test our defense mechanisms against C&W attacks on a black box attack assumption. For example, we generate adversarial examples using C0 and attack the Level One model C1. Under this black box assumption, we reduce the adversoral example successful rate by Level One model C1 to 18.63%, which was 85.0% under white box attack. It demonstrates that our defense mechanism is more effective under black box attacks. Future steps of the work will be on more investigation of defending black box attacks and improve defending white box attacks.

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
