# OpenReview forum: "Defending DNN Adversarial Attacks with Pruning and Logits Augmentation"
_ICLR.cc/2018/Workshop — Reject_

### Official Review · AnonReviewer1 · 2018-03-06

**Rating:** 4
**Confidence:** 4

**Review:**

The paper presents two approaches to makes models robust to adversarial attacks.
The first one consider pruning model weights similarly to what was proposed in Han et al (2015). The second one consider multiplying the weights of the last layer by a large constant (they call this "logits augmentation"). I don't understand how is this supposed to help. There's no mathematical explanation of what is happening when we do so.
Overall, results on adversarial examples seem better, but no results are given on clean examples (a sentence in the paper suggest results are not worse than the baseline, but a table would help).
Overall, two ideas are proposed but no explanation are given to why they should work, and how they compare to other known defenses.

---

### Official Review · AnonReviewer3 · 2018-03-09
**Pruning makes neural networks more robust**

**Rating:** 6
**Confidence:** 2

**Review:**

This paper presents two somewhat new ideas for defending against adversarial input attacks on image classification neural networks models.  They compress the model using the pruning technique from Han et. al. 2015, and they modify the weights in the last fully connected layer by a linear multiplier.  They show that the first idea on its own, can defend well against targeted FGSM and BIM on both MNIST and CIFAR10.  The combination of the two methods defends well against untargeted FGSM.  While neither of their techniques performs well against the attack from Carlini and Wagner (CW).

The paper has some grammatical problems, but is relatively easy to read and understand.

Pros
	- Work on a topic which is very hot right now
	- Two new defense mechanisms that can be added to (and possibly combined with) the existing tool chest of defenses
	- Strong performance against some adversaries

Cons:
	- neither idea is super novel.  The pruning idea is somewhat similar to defensive distillation and the logits augmentation is very similar to gradient inhibition from Liu 2018
	- they do not compare against existing defense mechanisms.  As the effectiveness of defenses depends on lot on detailed choices in the attack it's hard to know for sure how other defenses would perform in their setup, making it difficult to see how their technique fares

All in all, I think it's interesting that network pruning alone performs so well as a defense, and I think others in the community will find this result interesting as well, so I while  I think the paper is not super strong, I do believe it's above the bar for acceptance.

---

### Decision · Program_Chairs · 2018-03-20
**ICLR 2018 Workshop Acceptance Decision**

**Decision:**

Reject

**Comment:**

Based on the reviews, this paper has not been accepted for presentation at the ICLR workshop. However, the conversation and updates can continue to appear here on OpenReview.